# Economic Evaluation of Different Screening Strategies for Severe Combined Immunodeficiency Based on Real-Life Data

**DOI:** 10.3390/ijns7030060

**Published:** 2021-09-15

**Authors:** M. Elske van den Akker-van Marle, Maartje Blom, Mirjam van der Burg, Robbert G. M. Bredius, Catharina P. B. Van der Ploeg

**Affiliations:** 1Unit Medical Decision Making, Department of Biomedical Data Sciences, Leiden University Medical Center, P.O. Box 9600, 2300 RC Leiden, The Netherlands; 2Laboratory for Pediatric Immunology, Department of Pediatrics, Willem-Alexander Children’s Hospital, Leiden University Medical Center, P.O. Box 9600, 2300 RC Leiden, The Netherlands; m.blom@lumc.nl (M.B.); m.van_der_burg@lumc.nl (M.v.d.B.); 3Department of Pediatrics, Willem-Alexander Children’s Hospital, Leiden University Medical Center, P.O. Box 9600, 2300 RC Leiden, The Netherlands; R.G.M.Bredius@lumc.nl; 4Department of Child Health, The Netherlands Organization for Applied Scientific Research, TNO, P.O. Box 3005, 2301 DA Leiden, The Netherlands; kitty.vanderploeg@tno.nl

**Keywords:** newborn screening, severe combined immunodeficiency, cost analysis

## Abstract

Although several countries have adopted severe combined immunodeficiency (SCID) into their newborn screening (NBS) program, other countries are still in the decision process of adding this disorder in their program and finding the appropriate screening strategy. This decision may be influenced by the cost(-effectiveness) of these screening strategies. In this study, the cost(-effectiveness) of different NBS strategies for SCID was estimated based on real-life data from a prospective implementation study in the Netherlands. The cost of testing per child for SCID was estimated at EUR 6.36. The cost of diagnostics after screen-positive results was assessed to vary between EUR 985 and 8561 per child dependent on final diagnosis. Cost-effectiveness ratios varied from EUR 41,300 per QALY for the screening strategy with T-cell receptor excision circle (TREC) ≤ 6 copies/punch to EUR 44,100 for the screening strategy with a cut-off value of TREC ≤ 10 copies/punch. The analysis based on real-life data resulted in higher costs, and consequently in less favorable cost-effectiveness estimates than analyses based on hypothetical data, indicating the need for verifying model assumptions with real-life data. The comparison of different screening strategies suggest that strategies with a lower number of referrals, e.g., by distinguishing between urgent and less urgent referrals, are favorable from an economic perspective.

## 1. Introduction

Newborn screening (NBS) aims at detecting conditions shortly after birth that are treatable but not clinically evident in the newborn period. By detecting these conditions in an early phase, the clinical manifestation of the disease may be prevented, or the course of the disease might be influenced positively. NBS was first introduced in the United States in the early 1960s using screening cards with dried blood spots [1], and has expanded to countries around the world, while the number of conditions included in NBS programs is also growing.

One of the most life-threatening inherited disorders of the immune system is severe combined immunodeficiency (SCID). Patients with SCID are usually born asymptomatic but present with severe, recurrent infections, chronic diarrhea, and failure to thrive in the first months of life. Without curative treatment in the form of hematopoietic stem cell transplantation (HSCT) or gene therapy, a fatal outcome is inevitable [2]. Previous studies showed that the early detection and treatment of SCID patients in the pre-symptomatic phase is associated with improved outcomes and higher survival rates [3,4,5]. Particularly, an infection-free status at the time of HSCT is important, herewith highlighting the importance of early detection and protective management to prevent infections.

Early detection of SCID can be realized via NBS by the detection of T-cell receptor excision circles (TRECs) in dried blood spots with quantitative polymerase chain reaction (qPCR) [6,7]. TRECs are circular DNA fragments formed during the T-cell receptor gene rearrangement. The absence of TRECs is indicative of the absence of recently formed naïve T-cells. There is a range of neonatal conditions and disorders that can be associated with T-cell lymphopenia and low TRECs around birth that are not related to SCID. Low or absent TRECs can also be identified in preterm newborns, newborns with congenital malformation, or T-cell impairment syndromes [8,9]. These findings can be considered secondary findings of NBS for SCID. To distinguish SCID from other T-cell lymphopenias, follow-up diagnostics by flow cytometric immunophenotyping and genetic analysis are indicated.

Although several countries have adopted SCID into their NBS program, one of the issues that remains to be solved is finding the appropriate screening strategy that balances a high sensitivity and avoiding missing neonates with SCID, while preventing high referral rates and a high number of secondary findings. A high referral rate is associated with a high emotional impact for parents, high workloads for downstream referral centers, and high diagnostic costs. Therefore, decisions have to be made on the appropriate TREC cut-off value and screening algorithm.

Whereas some countries are optimizing their screening strategy for NBS for SCID, other countries are still in an ongoing discussion about the implementation of this disorder in their program. The decision of adding a disease to the NBS program may be influenced by a cost-effectiveness analysis in which the additional effects of screening are related to the costs compared to a situation without screening. In cost-effectiveness studies on adding SCID to the NBS program from the USA, New Zealand, and the UK, incremental cost-effectiveness ratios (iCER) ranged from EUR 19,000 to 44,000 per quality of life-year (QALY) gained [10,11,12,13,14]. A Dutch model study for a hypothetical cohort based on literature estimates and expert opinion resulted in an iCER of EUR 33,400 per QALY gained, suggesting that SCID screening in the Netherlands might be cost-effective, but pilot studies are warranted to reduce the uncertainty around the estimates [15].

In the Netherlands, a prospective implementation pilot study on NBS for SCID (SONNET-study) started in April 2018, with the aim to gather knowledge about the practical implications of NBS for SCID, test qualities, costs, and the perspective of users (i.e., health care providers and parents). In this study, the costs of screening and diagnostics for different NBS strategies for SCID were assessed based on real-life data from the prospective implementation study. Furthermore, the previously used model was updated with these data to explore the consequences for the estimates of the iCER of SCID screening compared to a situation without screening.

## 2. Materials and Methods

### 2.1. Prospective Implementation Pilot

For the SONNET-study, all parents of newborns born in three of the twelve provinces of the Netherlands (Utrecht, Gelderland, and Zuid-Holland) were asked to participate in a research project on NBS for SCID (opt-out consent). All dried blood spots (DBS) included were collected as part of the Dutch routine NBS program from April 2018 onward. The SONNET-study was approved by the Medical Ethics Committee of the Erasmus MC, University Medical Center, Rotterdam (MEC-2017-1146).

### 2.2. TREC Analysis

TREC analysis was performed according to the SPOT-it™ kit instructions for use (ImmunoIVD, Stockholm, Sweden) according to a preset screening algorithm [16]. NBS cards with TRECs below cut-off required repeated analysis in duplicate (retest). Full-term infants with repeated TREC levels below cut-off had an abnormal screening result and were referred for follow-up diagnostics. Preterm infants with abnormal results required a second specimen to be collected at the corrected gestational age of 37 weeks (second heel prick). Abnormal screening results with low β-actin levels were considered inconclusive and required repeated sampling (repeated first heel prick).

### 2.3. Adjusted Cut-Off Values and New Screening Algorithm (Post Hoc)

From April 2018 to October 2018, newborns with TREC ≤ 6 copies/3.2 mm punch were referred for clinical follow-up, according to the kit instructions of the manufacturer (ImmunoIVD). After six months of screening, the TREC cut-off value was increased to ≤10 copies/3.2 mm punch to ensure that no atypical SCID cases would be missed. For this study, the adjustment in cut-off value allows the investigation of both screening situations (cut-off TREC ≤ 6 and ≤10) as all newborns from November 2018 with TRECs ≤ 10 were referred for follow-up diagnostics. Screening data were included from 1 November 2018 to 31 July 2020 (*N* = 127,160 screened newborns).

There are a number of medical conditions without an intrinsic defect in the number of T-cells leading to low TREC levels around birth. Some of these conditions could resolve within the first few days to weeks after birth, leading to the normalization of TRECs levels. For this reason, a new screening algorithm was developed post hoc that distinguishes between urgent referrals with TREC levels ≤ 2 copies/3.2 punch and cases with TREC levels > 2 to ≤10 that require a second heel prick after seven days. Based on retrospective data of the SONNET-study, it was determined which newborns would be directly referred and which newborns would have required a second heel prick [17].

### 2.4. Cost of Screening

Screening data from the SONNET-study were obtained via the NEONAT database, the national laboratory information system in which all NBS test results are stored. The numbers of first heel pricks, duplicate analyses, and second duplicate analyses (all on the first blood sample) were obtained, as well as the numbers of repeated first heel pricks and second heel pricks needed and performed (and duplicate analyses in these). For the new post hoc screening algorithm, the number of additional second heel pricks was determined based on the number of children with TREC > 2 to ≤10. In the cost calculations, we assumed all would have been performed.

Costs of the screening test were assessed using the microcosting approach, by collecting detailed data on resources utilized and the value of those resources [18]. The price level of 2020 was used (2020 euros). Cost of screening consists of costs of the TREC assay, use of laboratory equipment, and material and personnel. Costs of the assay were based on the arrangement between the manufacturer and Dutch screening laboratories. These included the lease of the thermal cycler and qPCR instrument. Costs of other equipment were obtained by the straight-line depreciation of the equipment needed in each of the five screening laboratories in the Netherlands, assuming a lifetime of 5 years, maintenance costs and interest, and a nationwide use of 170,000 times a year [19]. In addition, the yearly cost of additional laboratory personnel (laboratory technician 0.6 fte/lab, scientific staff 0.1 fte/lab, 5 labs) needed for SCID screening was divided by the yearly number of SCID tests, to obtain personnel costs per test. The cost of blood collection and logistics were not included for the first test, as heel prick blood samples are already processed for other screening purposes. In the case a repeated first heel prick or a second heel prick sample is needed for SCID, the costs of blood collection and logistics were included.

In some cases, test results indicated the need for an additional heel prick, but this was not performed, e.g., because the child passed away before the heel prick could be performed. Costs were only accounted for when the heel prick was actually performed.

### 2.5. Cost of Diagnostics

Information on the diagnostic process of infants with an abnormal SCID screening result were obtained from the academic hospitals participating in the SONNET-study. Numbers and types of tests and clinical contacts, outpatient visits, and hospital days were retrieved from the medical records of the children referred until November 2020. At that time, diagnostics were completed for the majority of the children. If not, the diagnostics that were expected to take place have been included in the analysis. Subsequently, health care use was multiplied with cost prices. Cost prices were obtained from the Dutch costing manual [19,20] and the Dutch Healthcare Authority [21].

To assess the cost of diagnostics in a situation without screening, a pediatrician (R.G.M.B) and clinical researcher (M.B.) reviewed the medical records of the infants referred and estimated which diagnostics would have likely happened in a situation without screening. Costs are reported in 2020 euros.

### 2.6. Cost-Effectiveness

The new estimates of costs of screening and diagnostics (in a situation with and without screening) and the number of children referred for the different screening strategies were included in the decision analysis model of Van der Ploeg et al. [15], to explore the consequences for the iCER of NBS for SCID compared to a situation without NBS for SCID. The decision analysis model used a lifetime horizon and employed the healthcare perspective. Model parameters are shown in Appendix A. A more detailed description of the model and sensitivity analyses is given elsewhere [15].

## 3. Results

Costs of testing on first heel pricks were determined at EUR 6.36 per heel prick card. Costs of repeat first heel pricks and second heel pricks were estimated at EUR 79.03 (see Table 1). Costs to refer a child were EUR 145 (1 h of work for a medical advisor at an hourly rate of EUR 145).

In the period from 1 November 2018 to 31 July 2020 127, 160 newborns were screened for SCID. Percentages of repeat first heel pricks and second heel pricks ranged between 0.003% and 0.006%, and 0.016% and 0.061%, respectively, for the different screening strategies. Costs of screening per newborn are comparable for the different screening strategies (see Table 2).

Fifty-six newborns obtained a positive screening result in the SONNET-study. None of them had a family history of SCID or was diagnosed in utero. Referral rates of the different screening strategies varied between 0.022% and 0.041%. Most of the referred newborns had secondary T-cell impairment.

Diagnostics after the positive screening test consisted of personnel time during clinical contacts, during initial hospital stay, outpatient visits, consultations, emergency care visits, additional hospital stays, and diagnostic tests such as flow cytometry and whole-exome sequencing using a whole-exome sequencing SCID gene panel (WES SCID). The average cost per screen-positive for the diagnostic procedures depends on the final diagnosis and ranged from EUR 985 to 8561 (see Table 3).

In a situation without screening, costs of diagnosis of SCID were assumed to be the same as in a situation with screening (EUR 7517). Cost of secondary T cell impairment, idiopathic lymphocytopenia, and T-cell impairment syndromes were assessed to be lower, EUR 486, 2250, and 5111, respectively. Logically, there are no costs for diagnostics after false-positive screen results in a situation without screening.

Including the observed cost for screening and diagnostics in the SCID model of Van der Ploeg et al. [15] (see Appendix A for comparison of new and old parameter estimates) results in comparable iCERs of EUR 41,300 per QALY for the screening strategy with TREC ≤ 6 copies/3.2 mm punch and EUR 41,600 per QALY for the new screening strategy. The screening strategy with TREC ≤ 10 copies/3.2 mm punch has a less favorable iCER of EUR 44,100 per QALY (see Table 4).

## 4. Discussion

In this study, the real-life costs of testing for SCID and diagnostics after a positive screening test were used for comparing the costs of different screening strategies for SCID, and performing a model-based exploration of the cost-effectiveness of the different screening strategies.

The cost of testing per child for SCID on heel prick blood was estimated at EUR 6.36, mainly consisting of the cost of the assay and some additional costs for equipment, personnel, and material. These costs are at the upper side of the range from EUR 3.50 to 6.79 of the cost of testing reported in the literature [10,11,12,14,15,22]. Only the upper value of EUR 6.79 reported by Clement et al. [22] when assuming dedicated equipment use, i.e., using equipment exclusively for TREC analyses, was higher. The study by Clement was also based on a microcosting study with real-life pilot data comparable to our study, which may be more reliable than the hypothesized costs in other studies. Using other assays, e.g., in-house methods, may lead to lower costs, but Dutch screening laboratories have strict criteria for accreditation and, therefore, a CE-IVD-marked assay is preferred. Furthermore, the high cost of screening is also due to the fact that SCID screening is the first PCR-based test in neonatal screening programs. The implementation of this relatively new assay is associated with cost for extra equipment, reagents, and extra personnel. When in a later phase, other conditions such as spinal muscular atrophy (SMA) will be added to the NBS program, and the TREC assay can be extended with additional primers/probes in a multiplex setting. This implies that with limited extra reagent costs, screening for additional condition(s) will become possible. This will be relatively favorable for the incremental cost-effectiveness of these programs.

Referral rates between the screening strategies evaluated in this study varied between 0.022% and 0.041%. This is comparable to other modeling [10,11,12,13,14] except for our previous study in which a referral rate of 0.08% was used [15], based on referral rates found in a systematic review on TREC-based screening for SCID [23].

Costs of diagnostics after screen-positive results based on real-life data were assessed to vary between EUR 985 for children with a false-positive test result to EUR 8561 for children finally diagnosed with idiopathic lymphocytopenia. Flow cytometry and whole-exome sequencing with a SCID filter (WES SCID) were the major cost drivers. However, in a situation without screening, part of these costs will also occur.

In addition, the costs of diagnostics of screen-positive children assumed in other studies were quite low compared to our real-life estimates. Some studies have only assumed the need of a single appointment with a diagnostic test (flow cytometry) of EUR 209 [11,12] for screen-positive infants. McGhee et al. [13] also added T cell proliferation assays, which resulted in an amount of EUR 385. Next to assuming these presumptive positive costs consisting of an appointment and flow cytometry of EUR 276, Bessey et al. [10] distinguished by cost between diagnosis: the costs of SCID diagnosis were assumed to be EUR 819 (appointment and genetic testing) and the cost of diagnosis of idiopathic SCID and syndromes to be EUR 1786 (appointment and genetic testing 206 exome panel). Van der Ploeg et al. [15] assumed the cost of diagnostics to be EUR 1598, consisting of an appointment, flow cytometry, visit to pediatrician, repeat flow cytometry for 2/3 of screen-positives, and genetic test for 1/3. Apparently, the diagnostic procedure in practice consists of more testing and clinical care than was theoretically thought. This may be due to the fact that hypothetical costs are not realistic, e.g., flow cytometry and genetic testing ask for at least two appointments, one for explaining the test and obtaining the blood sample and one for discussing the test result (which may be performed by phone), while, in most studies, only one appointment is mentioned. In addition, in practice, more testing and clinical care might be performed than included in the protocols for diagnostics after a positive screening test. Last, as the whole procedure was new for care providers during this pilot study, extra diagnostic tests might have been requested to ascertain that no diagnoses were missed.

However, also in a situation without NBS, diagnostic costs will be made for part of the screen-positive infants. Comparing estimates for these costs for this study population to the observed diagnostic costs, additional costs of diagnostics due to screening were estimated to be EUR 1061, 6409, and 1492 for children with secondary T cell impairment, idiopathic lymphocytopenia, and T-cell impairment syndromes, respectively. Furthermore, in some of these earlier non-SCID diagnoses, a longer diagnostic trajectory may have been avoided and earlier treatment was enhanced, which may have led to cost savings and additional health benefits not included in this analysis. In addition, our assumption that diagnostic costs for SCID are comparable in a situation with and without NBS might be conservative, as, in a situation without NBS, more testing might be needed to discover that the symptoms are caused by SCID.

Our real-life study leads to higher iCERs mainly due to the higher screening costs, varying from EUR 41,300 per QALY for the screening strategy with TREC ≤ 6 copies/3.2 mm punch and EUR 41,600 per QALY for the new screening strategy, to EUR 44,100 for the screening strategy with a cut-off of TREC ≤ 10 copies/3.2 mm, compared to our earlier estimate of EUR 33,400 per QALY based on literature data and expert opinions [14], and to most of the other cost-effectiveness studies based on existing data, literature estimates, and expert opinions, where iCERs are mainly in the range of EUR 19,000-29,000 per (quality-adjusted) life year gained [10,11,12,14]. Only one study from 2005 reported an iCER of EUR 44,000 per QALY gained [13].

These higher iCERs obtained with real-life data are still in the range of the willingness to pay (WTP) values of EUR 20,000 to 80,000 per QALY that are considered acceptable in the Netherlands [24].

From an economic point of view, a screening strategy with a cutoff of TREC ≤ 10 copies/3.2 mm is not preferred, while outcomes for a screening strategy with a cutoff of TREC ≤ 6 copies/3.2 mm punch and the new screening algorithm are comparable. However, the new screening algorithm distinguished by urgent referrals for TREC levels 0–2 copies/punch and repeat heel pricks for cases with TREC levels > 2 to ≤10 resulted in the lowest number of referrals, thereby preventing emotional stress for parents [16] and workloads for downstream referral centers, which may be arguments to prefer this screening algorithm. It is worth considering second-tier test options that can reduce the number of referrals even more, although a second-tier test does come with extra costs [17].

This study also has some limitations. Due to the small-scale nature of pilot studies and the low referral rates, the numbers of children with screen-positive results are relatively low in this study, resulting in an uncertainty around our estimates of diagnostic costs. As real-life estimates appeared to differ clearly from hypothetical estimates, further research is needed based on larger cohorts. Furthermore, the costs and effects of the new screening algorithm were partly based on assumptions. These assumptions have to be confirmed using real-life data. This will be possible in the future, as this screening algorithm is used in the Netherlands from January 2021 onwards. Finally, in the explorations on cost-effectiveness, assumptions about the situation without screening and (long-term) treatment costs and effects were still based on literature data and expert opinions. Future research is needed in which real-life estimates for these items are obtained, as this may also influence the estimates on cost-effectiveness.

In conclusion, our analysis based on real-life data results in higher costs of screening and diagnostics, and consequently in less favorable cost-effectiveness estimates for NBS for SCID than those reported in previously published analyses based on hypothetical data, indicating the need for verifying model assumptions with real-life data. Comparisons of different screening strategies suggest that strategies with a lower number of referrals, e.g., by distinguishing between urgent and less urgent referrals, are favorable from an economic perspective.

## Figures and Tables

**Table 1 IJNS-07-00060-t001:** Cost of first, repeated first, and second heel prick for severe combined immunodeficiency (SCID) in 2020 euros.

Cost Item	First Heel Prick	Repeated First Heel Prick/Second Heel Prick
Blood collection	- ^2^	€22.05
Postage cost	- ^2^	€0.92
Sample processing	- ^2^	€2.70
Administration	- ^2^	€47.00 ^3^
Testing ^1^	€4.94	€4.94
Other equipment	€0.11	€0.11
Laboratory personnel	€1.28	€1.28
Materials	€0.03	€0.03
Total costs	€6.36	€79.03

^1^ including 21% value added tax (VAT), for the T-cell receptor excision circle (TREC)-assay, as well as use of the thermal cycler and quantitative polymerase chain reaction (qPCR) instrument, and including costs for retest of the same sample of newborn screening (NBS) card. ^2^ no additional costs compared to existing heel prick screening. ^3^ 0.5 h of work at an hourly rate of EUR 94.

**Table 2 IJNS-07-00060-t002:** Number of 1st heel pricks, repeated 1st heel pricks, 2nd heel pricks, and referrals and cost of screening based on SONNET trial in the period 1 November 2018 to 31 July 2020 (in 2020 euros).

Screening Strategy	TREC ≤ 6 Copies/3.2 mm Punch		TREC ≤ 10 Copies/3.2 mm Punch		New Screening Algorithm ^1^	
	# (% of FHP)	EUR	# (% of FHP)	EUR	# (% of FHP)	EUR
First heel pricks (FHP)	127,160	808,367	127,160	808,367	127,160	808,367
Repeated first heel pricks	4 (0.003%)	316	8 (0.006%)	632	8 (0.006%)	632
Second heel pricks	20 (0.016%)	1581	35 (0.028%)	2766	77 ^2^ (0.061%)	6085
Referrals	33 (0.026%)	4785	52 (0.041%)	7540	28 ^3^ (0.022%)	4060
- SCID	1		1		1	
- Secondary T-cell impairment	18		29		14	
- Idiopathic lymphocytopenia	3		6		4	
- T-cell impairment syndromes	7		9		5	
- False-positive	4		7		4	
Total costs		815,048		819,305		819,144
Cost per newborn screened		6.41		6.44		6.44

^1^ Direct referral if TREC levels ≤ 2 copies/3.2 punch, and cases with TREC-levels > 2 to ≤10 require a second heel prick after seven days. ^2^ For the new post hoc screening algorithm, the number of additional second heel pricks was determined based on the number of children with TREC > 2 to ≤10 (*n* = 40). In the cost calculations, we assumed all would have been performed. ^3^ Number of referrals extrapolated. Data from the SONNET-study showed that 12 out of 52 referrals (23%) would have been directly referred for follow-up diagnostics (TREC 0–2), while 40 out of 52 referrals (77%) would have required a second NBS card (TREC > 2 to ≤10) with the new screening algorithm. Of these 40 referrals, peripheral blood cards spotted at the time of flow cytometry (approximately one week after first DBS) were available for 26 referred newborns. These were used as if they were the outcomes of a second heel prick. For the missing 14 blood samples, outcomes of the 26 available cards were extrapolated per diagnosis or diagnostic category.

**Table 3 IJNS-07-00060-t003:** Average costs per child of diagnostic procedures and clinical care in screen-positive newborns in SONNET-study in the period 1 November 2018 to 31 July 2020 (in 2020 euros).

	SCID	Secondary T-Cell Impairment	Idiopathic Lymphocytopenia	T-Cell Impairment Syndromes	False-Positives
	(*n* = 1)	(*n* = 33)	(*n* = 6)	(*n* = 9)	(*n* = 7)
Diagnostic procedures					
- Flow cytometry	472	719	1542	1277	612
- Whole-exome sequencing (WES)	5459	496	5459	1213	0
- Other diagnostics	591	131	626	784	112
Total diagnostic procedures	6521	1346	7626	3274	724
- Clinical care					
- Clinical contacts	0	15	0	7	0
- Outpatient visits	775	94	535	320	158
- Phone consults	221	40	332	197	103
- Emergency care	0	0	47	95	0
- Consultations	0	9	21	0	0
- Additional hospital stay	0	43	0	2580	0
Total clinical care	996	201	935	3198	261
Total	7517	1547	8561	6473	985
(min, max)	(7517, 7517)	(0, 7756) ^1^	(6603, 11,480)	(253, 23,628)	(655, 2024)

^1^ Some infants died shortly after referral, before diagnostics started.

**Table 4 IJNS-07-00060-t004:** Model-based yearly cost and effects per 100,000 infants in a situation with and without newborn screening for SCID, and incremental cost-effectiveness ratios for different screening strategies (in 2020 euros).

	TREC ≤ 6 Copies/ 3.2 mm Punch	TREC ≤ 10 Copies/ 3.2 mm Punch	New Screening Algorithm	No Screening
Costs of screening and	671,600	703,500	674,100	-
additional diagnostics ^1^				
- Screening	641,000	644,300	644,100	-
- Additional diagnostics ^1^	30,600	59,200	30,000	-
Cost of SCID treatment	269,000	269,000	269,000	456,400
Total healthcare costs	940,600	972,500	943,100	456,400
Number of children with SCID detected				
- early	1.72	1.72	1.72	0.38
- late	0	0	0	1.34
QALYs gained ^1^	11.7	11.7	11.7	0
Cost per QALY gained ^1^	41,300	44,100	41,600	-

^1^ Compared to a situation without screening.

## Data Availability

The data presented in this study are available on request from the corresponding author. The data are not publicly available, due to privacy.

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
