# Peer review of "Economic Evaluation of Different Screening Strategies for Severe Combined Immunodeficiency Based on Real-Life Data"

_2409-515X, 2021, doi:10.3390/ijns7030060_

Round 1

Reviewer 1 Report

Undertaking a  study based economic evaluation of SCID screening is important to convince policy makers of adding this test to the current newborn testing assay. I am pleased to see that the Netherlands undertook this study, not unlike the one done in France for the same purpose. The reference of our study should be updated (authors' ref 22): 

Clinical and economic aspects of newborn screening for severe combined immunodeficiency: DEPISTREC study results. Thomas C, Durand-Zaleski I, Frenkiel J, Mirallié S, Léger A, Cheillan D, Picard C, Mahlaoui N, Riche VP, Roussey M, Sébille V, Rabetrano H, Dert C, Fischer A, Audrain M.Clin Immunol. 2019 May;202:33-39. doi: 10.1016/j.clim.2019.03.012. Epub 2019 Apr 1.PMID: 30946917   The Dutch study followed an appropriate methodology. I have the following suggestions for the authors:   1) Among the cases detected, were there any newborn with a family history for whom the screening was therefore of limited use and was there any case of in utero treatment? if either please just mention it.   2) I am not sure that the extrapolation of the results is really necessary. In my view, it rather weakens the study by adding much uncertainty to otherwise robust results. This is just my opinion, I understand that the results look better  with an ICER, however I would suggest that the authors stop at the comparison between early and late detection and discuss the benefits of the former.  In our DEPISTREC study, there was a much higher cost in the detection arm just because one child had a rare mutation and required two HSCT which eventually failed. These very unfortunate events occur and the benefit of a real life study over a model is to demonstrate it.   Overall it is a very important piece of research that supports a policy change.      

It is a 

Author Response

We thank the reviewer for the time invested in the review and the useful comments. 

Point by point response

The reference of our study should be updated (authors' ref 22): Clinical and economic aspects of newborn screening for severe combined immunodeficiency: DEPISTREC study results. Thomas C, Durand-Zaleski I, Frenkiel J, Mirallié S, Léger A, Cheillan D, Picard C, Mahlaoui N, Riche VP, Roussey M, Sébille V, Rabetrano H, Dert C, Fischer A, Audrain M. Clin Immunol. 2019 May;202:33-39. doi: 10.1016/j.clim.2019.03.012. Epub 2019 Apr 1.PMID: 30946917  

We thank the reviewer for this update, and changed the reference.

Among the cases detected, were there any newborn with a family history for whom the screening was therefore of limited use and was there any case of in utero treatment? if either please just mention it.  

There were no newborns with a family history nor any in utero diagnosis among the cases detected. We added this information in the text (page 5, line 194-196): Fifty-six newborns obtained a positive screening result in the SONNET study. None of them had a family history of SCID or was diagnosed in utero.

I am not sure that the extrapolation of the results is really necessary. In my view, it rather weakens the study by adding much uncertainty to otherwise robust results. This is just my opinion, I understand that the results look better  with an ICER, however I would suggest that the authors stop at the comparison between early and late detection and discuss the benefits of the former.  In our DEPISTREC study, there was a much higher cost in the detection arm just because one child had a rare mutation and required two HSCT which eventually failed. These very unfortunate events occur and the benefit of a real life study over a model is to demonstrate it. Overall it is a very important piece of research that supports a policy change.      

We agree with the reviewer on the benefits of a real life study, which is the reason that we performed our real life analysis. The exploration of the effects of using these real life data is only a bonus, however with an important message for modelers by indicating the need for verifying model assumptions with real-life data as far as possible.

Reviewer 2 Report

The article is quite confusing due to its presentation:

  1. Is this paper advocating the use of real-life data as input parameters for models and using SCID screening strategies as an example or is the focus about updating the model using real-life data? If it's the latter, why is the title highlighting the source of the screening strategies ('real-life data') and why aren't there any brief description of the model? Would it also be useful then to provide comparison of results between real-life parameters used and those from literature reviews in the tables?
  2. This economic evaluation does not follow the CHEERS checklist which has been increasingly adopted by journals to optimise reporting of health economic evaluations. I would advocate using this checklist so that audiences can understand their work.
  3. The title is confusing as it suggests a descriptive cost analysis when it's more like a cost-effectiveness analysis (or cost-consequences analysis). If the authors wishes to make keep it generic, why not just use the term "economic evaluation" instead of "cost analysis"?
  4. Is it the incremental cost-effectiveness ratio you're looking at (C1-C0)/(E1-E0), or cost effectiveness ratio that you're looking at (C1/E1)?
  5. This is a modeling exercise that utilises real-life data to fit the model right? Can you please explain more about the model (in accordance to the CHEERS checklist)? Can you also explain why modeling was done even though there is real-life data please - why can't a within-study cost-effectiveness analysis be done?
  6. For the pilot, what are the time-points of follow-up?
  7. "Consequences for cost-effectiveness of SCID screening were explored." What does this mean? Do you mean cost-consequences of SCID screening were evaluated?
  8. Why isn't any sensitivity analysis done?
  9. Are there any willingness-to-pay threshold relevant to the Netherlands? How should a decision maker working in a screening committee in the Netherlands reading this article interpret the results?

Author Response

We very much appreciate the effort put into reading and critically reviewing our manuscript. The reviewer makes good points that we hope to have addressed satisfactorily in order to improve the manuscript.

Point-by-point response

  1. Is this paper advocating the use of real-life data as input parameters for models and using SCID screening strategies as an example or is the focus about updating the model using real-life data? If it's the latter, why is the title highlighting the source of the screening strategies ('real-life data') and why aren't there any brief description of the model? Would it also be useful then to provide comparison of results between real-life parameters used and those from literature reviews in the tables?

The focus of this article is on presenting real-life data on costs and diagnostics of SCID screening strategies. Furthermore, as a bonus we assessed the consequences of using this information instead of expert opinion as we did in our previous paper on the incremental cost-effectiveness ratio of SCID screening strategies compared to a situation without screening. We added a brief description of the model (page 4, line 169-172). Furthermore, we included an Appendix with the real-life parameters used in this paper and the parameters based on literature used in the previous study.  

  1. This economic evaluation does not follow the CHEERS checklist which has been increasingly adopted by journals to optimise reporting of health economic evaluations. I would advocate using this checklist so that audiences can understand their work.

As the model outcomes are just a bonus in this article, we decided to not include too much detail in this article on the model, which is extensively described in our previous paper, but to refer to our previous study. Therefore all information asked by the CHEERS checklist can be found in our previous paper (Van der Ploeg et al. Cost-effectiveness of newborn screening for severe combined immunodeficiency. Eur J Pediatr 2019, 178, 721–729). However, we extended the information on the model in this paper on page 4, line 169-172 and added an Appendix with model parameters: The decision analysis model used a lifetime horizon and employed the healthcare perspective. Model parameters are shown in Appendix 1. A more detailed description of the model and sensitivity analyses is given elsewhere [15].

  1. The title is confusing as it suggests a descriptive cost analysis when it's more like a cost-effectiveness analysis (or cost-consequences analysis). If the authors wishes to make keep it generic, why not just use the term "economic evaluation" instead of "cost analysis"?

We thank the reviewer for the suggestion to use the term economic evaluation and adapted the title in ‘Economic evaluation of different screening strategies for severe combined immunodeficiency based on real-life data’.

  1. Is it the incremental cost-effectiveness ratio you're looking at (C1-C0)/(E1-E0), or cost effectiveness ratio that you're looking at (C1/E1)?

We look at the incremental cost-effectiveness ratio of newborn screening for SCID compared to a situation without screening. We clarified this at several places in the manuscript.

  1. This is a modeling exercise that utilizes real-life data to fit the model right? Can you please explain more about the model (in accordance to the CHEERS checklist)? Can you also explain why modeling was done even though there is real-life data please - why can't a within-study cost-effectiveness analysis be done?

 As indicated above we extended the information on the model in this paper, and more detailed information can be found in our previous paper.

Newborn screening on SCID has lifelong consequences on cost and quality of life of children diagnosed. The additional costs of screening occur at the start of life, but it is hoped that these are more than compensated for by savings in cost and gains in quality of life during lifetime. Due to this lifetime horizon, no within study cost-effectiveness can be done. As we illustrated in this study combining real-life data and modelling might be the best way.

  1. For the pilot, what are the time-points of follow-up?

Diagnostics took place directly after a positive screening test. We included data on diagnostics until November 2020. For the majority of the patients the diagnostics were completed at that point. If not, the diagnostics that were expected to take place have been included in the analysis. We added this information on page 4, line 156-158: Numbers and types of tests and clinical contacts, outpatient visits and hospital days were retrieved from the medical records of the children referred until November 2020. At that time, diagnostics were completed for the majority of the newborns. If not, the diagnostics that were expected to take place have been included in the analysis.  

  1. "Consequences for cost-effectiveness of SCID screening were explored." What does this mean? Do you mean cost-consequences of SCID screening were evaluated?

We clarified the sentence as follows: Furthermore, the previously used model was updated with these data to explore the consequences for the estimates of the iCER of SCID screening compared to a situation without screening.

  1. Why isn't any sensitivity analysis done?

Multiple sensitivity analyses were performed in the previous study. Repeating this in the current study will not yield different results and insights.

  1. Are there any willingness-to-pay threshold relevant to the Netherlands? How should a decision maker working in a screening committee in the Netherlands reading this article interpret the results?

We added some information on willingness to pay threshold in the Netherlands (page 10, line 323-325): These higher iCERs obtained with real-life data are still in the range of the willingness to pay (WTP) values of 20,000 to 80,000 euros per QALY that are considered acceptable in the Netherlands [24].

Round 2

Reviewer 2 Report

The authors have adequately addressed my concerns and the content is a lot clearer now.